# The practice of defensive medicine among Jordanian physicians: A cross sectional study

Qosay Al-Balas[1]*, Shoroq Altawalbeh[2], Carmela Rinaldi[3,4], Ibtihal Ibrahim[5]

1 Department of Medicinal Chemistry and Pharmacognosy, Faculty of Pharmacy, Jordan University of Science & Technology, Irbid, Jordan, 2 Department of Clinical Pharmacy, Faculty of Pharmacy, Jordan University of Science and Technology, Irbid, Jordan, 3 Department of Translational Medicine, University of Piemonte Orientale, Novara, Italy, 4 Learning and Research Area, AOU Maggiore Della Carità, Novara, Italy, 5 Department of Pharmacy, Faculty of Pharmacy, Al-Zaytoonah University of Jordan, Amman, Jordan

* qabalas@just.edu.jo

## Abstract

### Background

Defensive medicine (DM) is a deviation from medical practice that is induced primarily by a threat of liability. While the DM behavior is well studied in the developed countries, little is known in developing countries and never been evaluated in Jordan.

### Objective

To evaluate the prevalence of DM practice in Jordan among physicians and to investigate reasons behind its practice and potential strategies to alleviate this practice.

### Methods

In this Cross-sectional study, self-administered questionnaire was distributed to a sample of physicians in both public and private sectors in Jordan. The collection period was from Jan 2021 to June 2021. The prevalence of DM practice was estimated among the study sample. Frequency scores of different DM behaviors, reasons of DM behaviors, and effectiveness of strategies in changing DM behaviors were summarized as average frequency scores with standard deviations. Multivariable linear regression models were conducted to evaluate potential predictors of total assurance and avoidance behavior scores.

### Results

A total of 175 Jordanian physicians completed the survey. The prevalence of adopting (or witnessing) DM behaviors among the study sample was 68% (n = 119). Diagnostic laboratory exams followed by prescribed medications were the most practiced behaviors in excessive rate during a typical working week. Unfavorable legislation for the physician was reported as the headmost reason for practicing DM, followed by pressure from the public and mass media opinion. Continuous update of knowledge, abilities, and performance and following specific protocols and/or appropriate clinical evidence and appropriate multidisciplinary and multi-professional communication were the most effective strategies that can mitigate DM behaviors.

**Data Availability Statement:** All relevant data are within the manuscript and its Supporting Information files.

**Funding:** The funding is received from Jordan University of Science & Technology, Deanship of

Research, Grant No.: 20200641. The funders had
no role in study design, data collection and
analysis, decision to publish, or preparation of the
manuscript.

**Competing interests:** The authors have declared
that no competing interests exist.

## Conclusions

Defensive medicine practice is common among Jordanian physicians with concerns about
increasing pattern in the future.

## Introduction

Medical malpractice has been debated intensively in literature and many definitions were
introduced. It can be defined as any act that is expressed by healthcare professionals which is
veered from medical norm and consequently engendered harm [1]. With the widespread
changes that have stroked all life sectors after the industrial revolution, physicians and medical
healthcare started to lose their paternalistic position and the care takers started to seek their
rights [2]. More choices are available for the patients, and only obeying what is dictated by
doctors is not becoming the norm anymore [3]. This has developed that patients started to
seek their privileges via litigation, a case that is hectic for the whole medical care system [4, 5].
Surely, the medical healthcare staff, especially physicians have to pursue defensively to protect
their status and avoid reputation annihilation. This act has been termed "Defensive medicine"
[6].

Defensive medicine was conceived in the 50s of the last century as the physician's aberra-
tion from the expected medical practice with the intention to avert criticism, prosecution, and
reputation subversion [7, 8]. It is classified as positive "assurance" or negative "avoidance" [8,
9]. Assurance implies needless diagnostic tests, medicines, and procedures. On the contrary,
avoidance deprives patients from benefits that require risky procedures or hospital admission
[10]. Both are confirmed to boost the expenditure of the offered health care to the patients and
unfortunately undermine the offered health services [9–11]. According to statistics in the US,
the estimated cost of defensive medicine in orthopedic surgery could be up to 2 billion USD
annually [12]. In Italy, 14% of the pharmaceuticals, 23% of laboratory test and 25% of the
imaging procedures increase ascribed to defensive medicine [13]. Litigation involving sports-
related spinal injuries in the US were analyzed and it was found that negligent supervision is
the most cause of legal litigation, followed by premises liability [14].

Defensive medicine is a disseminated phenomenon afflicting all medical branches, leading
to deficit in human, managerial, economic and organizational assets [15]. While the defensive
medicine behavior is well studied in the developed countries, little is known in developing
countries and never estimated in Jordan [12, 16, 17]. Jordan is a developing country with a
population of 10.5 million according to the Jordanian Department of Statistics for the year
2019 [18, 19]. Currently there are 110 hospitals of which 62 are private. This study aimed to
evaluate the prevalence of DM practice in Jordan among physicians and to investigate reasons
behind its practice and potential strategies to alleviate this practice.

## Methods

### Study design and population

This was a cross sectional study in which participants were physicians in both public and pri-
vate sectors in Jordan. A self-administered structured questionnaire was distributed in person
for a sample of physicians from both sectors. The collection period was from Jan 2021 to June
2021. Public sector hospitals included in the current study were hospitals belong to Royal Med-
ical Services (RMS) and Ministry of Health (MoH), in addition to King Abdullah University

Hospital (KAUH), a leading teaching hospital in Jordan. Physicians were informed verbally about the scope and aims of the study and were invited to participate in the study. Public sector hospitals included in the current study were King Abdullah University Hospital (KAUH), which is a teaching hospital, and government hospitals belong to Royal Medical Services (RMS) and Ministry of Health hospitals (MoH). This research was approved by the Institutional Review Board (IRB) Committee at KAUH (Ref number: 60/136/2020), and a verbal consent was taken from the participants in this study.

## Instrument and data collection

The current study was conducted using a validated published survey that was designed to describe the practice of DM, reasons for practicing DM and possible solutions to change DM behaviors among Italian hospital physicians [15]. The survey contains the following sections: (1) demographic information and professional practice characteristics; (2) personal experience/ opinion on the practice of DM in the medical profession, reasons for practicing DM as well as the impacts of practicing DM; (3) strategies aimed at changing DM behaviors and minimizing the practice of DM. Permission to use the survey was obtained from the corresponding author of the published manuscript [15]. The used survey in this work is reported in S1 File.

Although the survey was fully validated for the purpose of the study, practicality and clarity of the survey questions were evaluated through a pilot study of about 15 physicians. Feedback from pilot study participants were utilized to improve the questionnaire as needed. Rephrasing some terms was the main adaptation conducted on the original survey, however, the overall meaning was maintained. This is important considering the possible cultural differences between Jordanian and Italian physicians.

The current study evaluated the frequency of 6 assurance behaviors and 3 avoidance behaviors. The frequency score for each behavior was ranged from 0 to 10; 0 score indicates "the least frequent" and 10 score indicates "the most frequent". Total behavior frequency score was calculated for both assurance and avoidance behaviors for each patient by summing their frequency scores for each behavior with a maximum of 60 for assurance behaviors and 30 for avoidance behavior.

## Statistical analysis

Descriptive statistics (mean, standard deviation, frequency, and percentages) were used, as appropriate, to describe variables measured in the study. Participant characteristics were described as frequencies and percentages. Frequency scores of DM behaviors, reasons of DM behaviors, and effectiveness of strategies in changing DM behaviors were summarized as means with standard deviations for the whole sample and by the public and private sectors (provided in the supplementary material). Multivariable linear regression models were conducted to evaluate potential predictors of total assurance and avoidance behavior scores. All data analyses were conducted using Stata version 17 software (StataCorp. 2021. Stata: Release 17. Statistical Software. College Station, TX: StataCorp LLC.)

## Results

Table 1 shows the participant characteristics; a total of 175 physicians completed the questionnaire with response rate of 79.5%. Among the participated physicians, 73.71% were males while the age of 85.14% of the participants was less than 65 years old. About half of the participants in the sample were from the central region of Jordan 48.57%, whereas 51.43% were from the North. Regarding years of professional experience, 48% of physicians had more than ten years' experience. Among the working area, 54.29% of participants were from the private

**Table 1. Participant' characteristics.**

| | | Frequency | Percent |
|---|---|---|---|
| **Gender** | Female | 46 | 26.29 |
| | Male | 129 | 73.71 |
| **Age** | 24–40 | 92 | 52.57 |
| | 41–65 | 57 | 32.57 |
| | > 65 | 26 | 14.86 |
| **Region of work** | Middle | 85 | 48.57 |
| | North | 90 | 51.43 |
| **Years of professional experience excluding training years (internship)** | ≤1 | 7 | 4.00 |
| | 2–4 | 29 | 16.57 |
| | 5–10 | 55 | 31.43 |
| | >10 | 84 | 48.00 |
| **Present working setting** | Government | 74 | 42.29 |
| | Private | 95 | 54.29 |
| | University-based | 6 | 3.43 |
| **Physician specialty** | General medicine and primary care | 27 | 15.43 |
| | Surgery | 30 | 17.14 |
| | Gynecology | 25 | 14.29 |
| | Internal medicine | 24 | 13.71 |
| | Pediatrics | 14 | 8.00 |
| | Others | 55 | 31.40 |
| **Place of performing most professional activities** | Consulting | 30 | 17.10 |
| | Department | 60 | 34.30 |
| | E.R. | 36 | 20.60 |
| | Outpatient | 122 | 69.70 |
| **Number of patients the physician follows/visits in a typical working week (mean and SD)** | | 91.17 | 116.39 |
| **Number of instrumental examinations and/or diagnostic tests and/or laboratory examinations the physician performs on average in a typical working week (mean and SD)** | | 86.12 | 235.34 |

sector. Outpatient clinics were the place of performing most of the professional activities 69.70%. Surgeons comprised the largest specialty group among respondents 17.14%, followed by general medicine and primary care 15.43%. During the last year, 68% (119) of participants reported that they have had the opportunity to adopt (or witness) defensive medicine behaviors.

All the causes for practicing DM studied in this work achieved an average frequency score of more than five. Physicians who are working in (MoH or JMA) have ranked unfavorable legislation for the physician as the headmost reason for practicing DM (mean: 5.8, SD: 3.0032), followed by pressure from the public and mass media opinion (mean: 5.497, SD: 3.1511). However, the least frequent cause of DM behaviors was the low trust in the management (Company, hospital, etc.) (mean: 5.029, SD: 2.9035). Fig 1: shows more details related to the frequency of main causes of DM behaviors. Main causes of DM behaviors by sector are summarized in S1 Table.

Regarding frequency scores of DM behaviors, it was found that laboratory exams with diagnostic goals (mean: 3.857, SD: 3.1435), followed by drug prescription (mean: 3.794, SD: 3.3102) were the most frequent types of assurance the physicians practice as DM in excess if compared to real needs during a typical working week in Jordan. On the other hands, requests for professional activities (visits, instrumental exams, etc.) by patients (mean: 3.634, SD: 2.9052) followed by the staff avoids assisting a patient with high risk of complications (mean

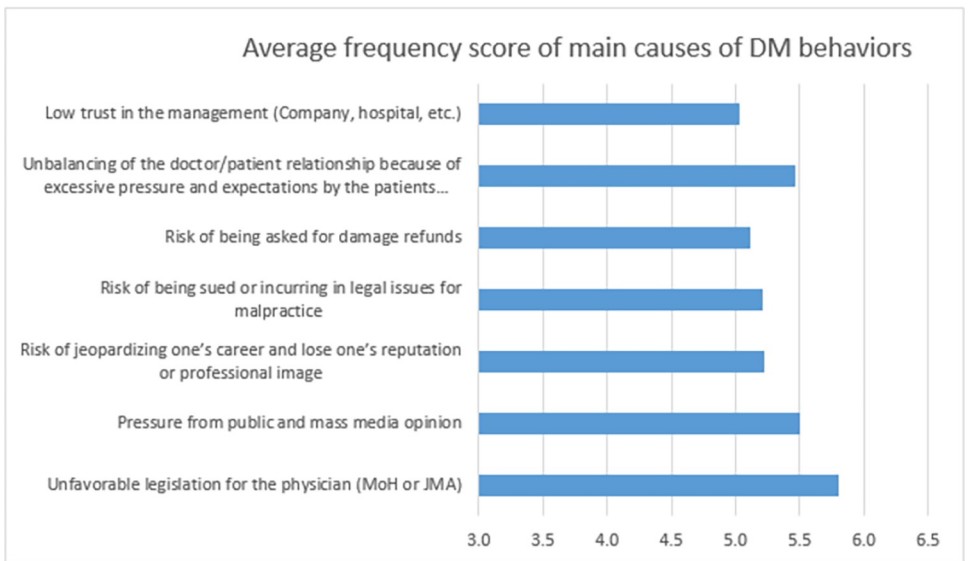

**Fig 1. Frequency of main causes of DM behaviors among physicians.** 0 score indicates "the least frequent" and 10 score indicates "the most frequent".

3.460, SD: 2.8256) were the most frequent types of avoidance DM behaviors performed potentially in excess if compared to real needs during a typical working week in Jordan. Table 2 summarizes average frequency score of DM behaviors during a typical working week. The average total scores for assurance and avoidance behaviors were 19.8 (SD = 15.8) and 10.5 (SD = 7.3), respectively. Average frequency score of different DM behaviors by sector are reported in S2 Table.

A total of 44 (25.1%) of the participants considered that the practice of defensive medicine is a favoring factor for the professional practice, whereas 99 (56.6%) participants considered that the practice of defensive medicine is a limiting factor for the professional practice score. Among potential reasons for considering DM as a favoring factor, the protection of DM

**Table 2. Average frequency score of DM behaviors during a typical working week.**

|  | Mean | SD |
|---|---|---|
| **Assurance behaviors** | | |
| Specialty consulting/ referrals | 3.029 | 2.7819 |
| Laboratory exams with diagnostic goals | 3.857 | 3.1435 |
| Instrumental examinations and other diagnostic tests | 3.463 | 3.0976 |
| Prescribed drugs | 3.794 | 3.3102 |
| ER referrals/hospital admissions | 2.845 | 3.0037 |
| Transfers to other departments/hospitals | 2.771 | 2.8896 |
| **Avoidance behaviors** | | |
| The staff avoids assisting a patient with high risk of complications | 3.460 | 2.8256 |
| The staff avoids performing potentially effective but high-risk treatments or procedures. | 3.366 | 2.7860 |
| Requests for professional activities (visits, instrumental exams, etc.) by patients that can be interpreted as defensive medicine | 3.634 | 2.9052 |

• 0 score indicates "the least frequent" and 10 score indicates "the most frequent"

• Participants were asked about their behaviors prescribed/performed compared to real needs

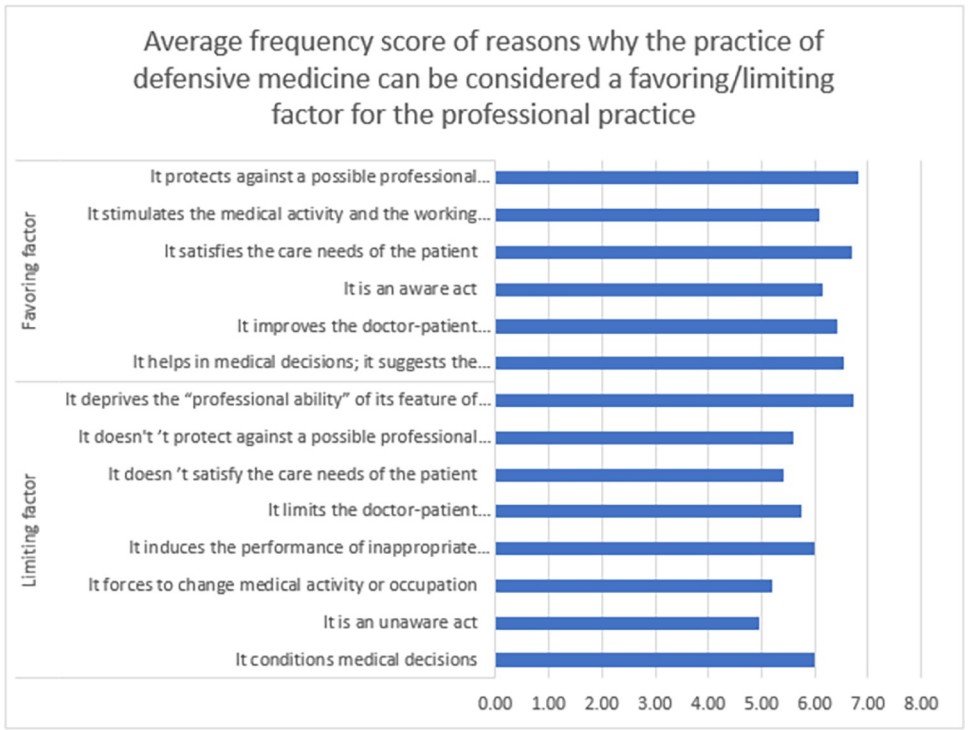

**Fig 2. Average frequency score of reasons why the practice of defensive medicine can be considered a favoring/limiting factor for the professional practice.** 0 score indicates "the least frequent" and 10 score indicates "the most frequent".

against a possible professional responsibility was the most frequent reason (average score: 6.818, SD: 2.7642), followed by satisfying the care needs of the patient (average score: 6.705, SD: 2.4360). While the most frequent reason for considering DM as a limiting factor was deprivation the professional ability and its feature of being the main reference for the patients (average score: 6.717, SD: 2.8322). Fig 2 reveals more details about the reasons why the practice of DM can be considered a favoring/limiting factor for the professional practice. Reasons why the practice of DM can be considered a favoring/limiting factor for the professional practice by sector are reported in S3 Table.

A total of 86 (49.1%) participants considered that the practice of DM is a favoring factor for the patients, whereas 59 (33.7%) participants considered that the practice of DM is a limiting factor for the patients. As depicted in Fig 3, the most frequent reported reason for considering DM as a favoring factor for the patients as believed by the physicians is that DM favors the perception of patient well-beings and satisfaction (increase in the perceived quality) (average score: 7.326, SD: 2.2199). However, the increase in waiting time was the most frequent reason for considering DM as a limiting factor for the patients as believed by the physicians (average score: 6.305, SD: 3.1473). Reasons why the practice of DM can be considered a favoring/limiting factor for the patients by sector are reported in S4 Table.

The suggested strategies to alleviate the practice of DM as perceived by physicians are evaluated in Table 3. Strategies were classified to personal and external actions. Continuous update of knowledge, abilities, and performance (average score: 8.39, SD: 2.20), following specific protocols and/or appropriate clinical evidence (average score: 8.32, SD: 2.29) and appropriate multidisciplinary and multi-professional communication (average score: 8.30, SD: 1.96) were the most frequent personal strategies from the physician's point of view. Regarding external

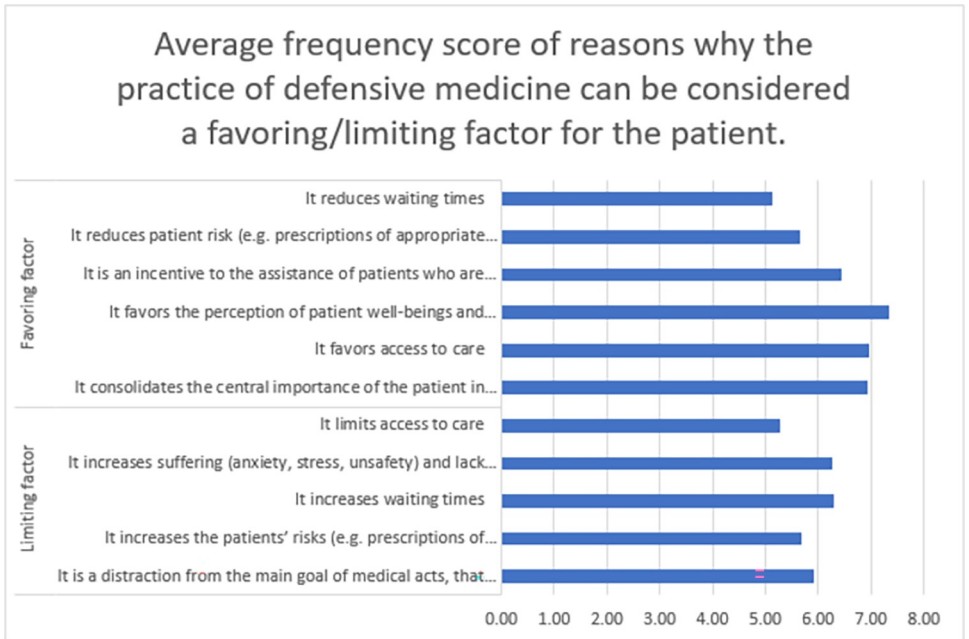

**Fig 3. Average frequency score of reasons why the practice of defensive medicine can be considered a favoring/limiting factor for the patients.** 0 score indicates "the least frequent" and 10 score indicates "the most frequent".

actions, reforming the regulations on professional responsibilities (average score: 7.31, SD: 2.73), incentives and/or professional rewards for positive medical performances (average score: 7.14, SD: 2.80) and greater interest of the public and mass media opinions in the healthcare activities that show value) (average score: 7.06, SD: 2.78) were the topmost external strategies.

The effectiveness score of different strategies in changing DM behaviors by sector are reported in S5 Table.

**Table 3. Average score of effectiveness of strategies in changing these defensive medicine behaviors.**

| | Mean | Std. Deviation |
|---|---|---|
| **Personal actions** | | |
| Follow specific protocols and/or appropriate clinical evidence | 8.32 | 2.29 |
| Continuously update knowledge, abilities, and performance | 8.39 | 2.20 |
| Appropriate multidisciplinary and multi-professional communication | 8.30 | 1.96 |
| Appropriate verbal and non-verbal communication with the patients | 7.87 | 2.31 |
| Adequate healthcare documents and updated medical diary | 8.00 | 2.30 |
| Participate to systematic and regular medical and clinical audits | 7.56 | 2.54 |
| Strengthen one's own ethical and professional values | 8.02 | 2.34 |
| Always report medical mistakes and participate to anonymous inquiries | 7.41 | 2.73 |
| **External actions** | | |
| Incentives and/or professional rewards for positive medical performances | 7.14 | 2.80 |
| Greater interest of the public and mass media opinions in the healthcare activities that show value (not only in real/hypothetical malpractice) | 7.06 | 2.78 |
| Greater support from the companies | 6.44 | 2.80 |
| Greater guidance by insurance companies | 5.98 | 3.02 |
| Reform of the regulations on professional responsibilities | 7.31 | 2.73 |

### Scores are ranked from 0 to 10 where 0 is "worst action" and 10 "the best action"

The majority of participants 153 (87.4%) believe that the adoption of DM behaviors increases the costs of healthcare interventions, 11 participants (6.3%) believe that it decreases the cost, and 11 participants (6.3%) believe that it does not change the costs.

Regarding the likelihood of incurring in legal issues, 73 (41.7%) participants believe that the adoption of DM behaviors decreases the likelihood, 52 participants (29.7%) believe that it increases the likelihood, and 50 participants (28.6%) believe that it does not change the likelihood.

As viewed by participants, 74.3% expect that DM behaviors are going to increase in the mid-term, whereas only 14.9% expect that they are going to decrease.

Working in a university-based hospital or in a private hospital was associated with lower frequency of assurance behaviors when compared to government hospitals; coefficient was (-15.5) and (-6.6) respectively. Table 4 shows more details about the predictors of total assurance and avoidance behavior scores.

**Table 4. Predictors of total assurance and avoidance behavior scores.**

| | Coefficient | P value | 95% confidence interval | |
|---|---|---|---|---|
| **Predictors of Assurance behaviors** | | | | |
| **Male gender** | 0.68 | 0.808 | -4.884 | 6.258 |
| **Age** | | | | |
| 24–40 | ref | | | |
| 41–65 | 2.58 | 0.505 | -5.050 | 10.208 |
| > 65 | -3.51 | 0.475 | -13.168 | 6.156 |
| **Years of professional experience** | | | | |
| >10 | ref | | | |
| 5 to 10 | 1.85 | 0.641 | -5.959 | 9.650 |
| 2 to 4 | 5.01 | 0.300 | -4.501 | 14.526 |
| ≤1 | -4.66 | 0.511 | -18.624 | 9.310 |
| **Present working setting** | | | | |
| Government | ref | | | |
| Private | -6.63 | 0.019 | -12.137 | -1.122 |
| University-based | -15.47 | 0.024 | -28.835 | -2.101 |
| **Number of patients the physician follows/visits in a typical working week** | 0.010 | 0.367 | -.0125 | .0336 |
| **Predictors of Avoidance behaviors** | | | | |
| **Male gender** | 0.916 | 0.475 | -1.612 | 3.444 |
| **Age** | | | | |
| 24–40 | ref | | | |
| 41–65 | 2.79 | 0.117 | -0.702 | 6.288 |
| > 65 | -1.82 | 0.418 | -6.245 | 2.606 |
| **Years of professional experience** | | | | |
| >10 | ref | | | |
| 5 to 10 | 1.72 | 0.343 | -1.852 | 5.296 |
| 2 to 4 | 3.09 | 0.162 | -1.249 | 7.419 |
| ≤1 | -4.16 | 0.201 | -10.561 | 2.234 |
| **Present working setting** | | | | |
| Government | ref | | | |
| Private | -1.61 | 0.208 | -4.127 | 0.906 |
| University-based | -2.53 | 0.414 | -8.655 | 3.577 |
| **Number of patients the physician follows/visits in a typical working week** | 0.007 | 0.175 | -0.003 | 0.0179 |

## Discussion

Defensive medicine scope in developing countries is lacking, a scenario that is quite different in developed countries. Jordan is a developing country with limited resources; however, it suffers as other countries in the world from this global phenomenon [20]. This study is considered the first ever study in Jordan that is oriented to physicians to obtain their views about DM practice. Mainly, this study seeks the physicians view about DM and the extend of its spread and to unravel the reasons behind such a flawed practice.

Findings from this study shows a spread of DM practice among physicians with a 68% of the sample surveyed. While this percentage implies a high rate of DM practice, it is far less compared to the US and Japan studies; it was up to 90% [1–3]. In addition, areas such as obstetric anesthesia was found to be the main source of malpractice liability [21, 22]. This difference could be attributed to the nature of the sample taken, the sensitivity of the working place (Army hospitals vs others), variety specialties, and the level of freedom the physician could have. Moreover, there is no distinct border about what is appropriate or inappropriate in clinical practice [2].

In concordance with the literature, Jordanian physicians emphasize that the main reason for practicing such behavior is troublesome legal legislations that has been enacted since the beginning of this century and being modified in new medical responsibility law at 2018 [2, 5, 10]. Consequently, DM is considered the secure sanctuary that physicians seek to avoid being sued by affected patients. As a matter of fact, the increased expectations of the patient in addition to the improved medical care practices and media could be a crucial reason for increasing such litigations. Thereupon, Jordanian physicians as well their international counterparts are seeking a refuge from being prosecuted or being their reputation is questionable [11]. Scenario that is being flipped when the physicians require career protection from the patients.

Expectedly, laboratory exams for diagnostic purposes were found to be the most practiced behavior by Jordanian physicians in excessive rate during a typical working week. This is in consonance with other studies such as the one performed in the UK in which 59% of the physician were ordering unnecessary laboratory test [5]. Other studies have ranked the expensive diagnostic imaging to be the most ordered procedure especially for orthopedic surgery specialty. The ascribed reason for such attitude is that these procedures would help proper and more accurate clinical judgments for the sake of the patient [12].

The second most assurance behavior was medicine prescribing. Although this result is not complying the one performed in Italy, this means that there should be more interference from the clinical pharmacists to cope with such condition and control drug prescription as excess medication means more side effects and more health deterioration of patients [4]. DM deprives the professional ability and its feature of being the main reference for the patients. So, the government should obligate physicians to restrict their practice based on unbiased, updated guidelines. It is obvious that DM is not a cost-effective practice [9].

Surprisingly, the avoidance conduct is approximately resembling the assurance practice in the current study. Previous studies showed lower prevalence of avoidance DM compared to assurance behaviors [8, 10–13]. This may be explained again of the fear of litigations especially the majority (17.14%) of the participants in this sample are from the general surgeon's specialty in which the consequences of the medical intervention might be more serious. This avoidance behavior in Jordan contradicts the international behavior especially in high-risk specialties [8, 14–17].

Approximately half of the physician have confirmed that the DM is a favoring factor for the patients, a description that foster the paternalistic view of the caregivers. Rather, futile medicine prescription and extra procedures are exposing the patient's life to danger above the extra

costs that could be incurred. It should be emphasized that the belief that more advanced and costly procedures not necessarily results in better patient care. Instead, proper treatment guidelines could be the exit form such situations [13].

One of the serious concern this study is revealing is that more than half of the surveyed physicians acknowledge that DM is a limiting factor for the professional practice. This can be overcomed partially via adopting and disseminating clinical guidelines and quality improvement initiatives supported by evidence-based studies. In addition, the other side of the equation (patient) should receive proper and structured education about these procedures [8, 14, 18].

According to the physician's point of view, DM practice could be alleviated and lessened in the future by subjecting the physicians to continuous and state of the art training courses and conferences to make sure that their knowledge still as up to date. New international guidelines should be delivered to them while updating any rules and regulations that could help protect physician but at the same time protect the patients' rights. This finding is in agreement with what is reported in the literature as recommendations to avoid DM practice [8, 14, 18].

This study results are in reconciliation with literature trend that DM conduct is imposing a financial burden on health systems of the countries. It depletes the countries resources; human, technical, social ones especially when the country's economy on the brink such as Jordan. Physicians in this survey warns about the catastrophic consequences of such phenomenon as 74.3% expected the DM will increase in the mid-term period the costs and will be considered as a thorny problem. This will further increase drug expenditure. The need for guidance for transparency in providing care emerged, subsequently, prevent or minimize the practice of DM.

Working in a university-based hospitals or private hospitals was associated with lower frequency of assurance behaviors when compared to public hospitals. Physicians in a university-based hospital are more in touch with updated guidelines due to their academic affiliation and continuous education. Again, this supports the importance of investing in physician's knowledge of guidelines and continuous education. Also, being treated in a private hospital makes the costs of treatments, tests and procedures a concern for both physicians and patients because of reimbursement issues compared to public hospitals in which most of these treatments and procedures are extensively covered by government. The wide and extensive coverage in the public sector in Jordan is definitely a predisposing factor for assurance DM practice that needs further attention.

Importantly, it deserves to mention the limitations this study has suffered from. One major criterion is the absence of clear distinction about the definition of proper practice and the improper one [18]. Moreover, it is challenging to disentangle liability-related motivational factors from other factors that influence clinical decision-making, such as physicians' general desire to meet patients' expectations and preserve trust [19]. In addition, these results are representing two areas of three ones in Jordan, as the southern area still is underrepresented. One major limitation in the current study is the sample size that was achieved through convenient sampling. Work settings and COVID-19 restrictions were challenging in terms of obtaining more responses during the study period. In addition, discussing the DM concept was not vey plausible or attractive for many of the encountered physicians. Finally, social-desirability bias might contribute to underestimating the actual prevalence of DM behaviors in the current study.

## Conclusion

Defensive medicine is common among practicing physicians in Jordan. Ordering unnecessary laboratory tests and prescribed medications are the most popular DM behaviors. Efforts to

decrease DM practice should focus on developing and disseminating clinical guidelines and provide an extensive education for physicians regarding appropriate care in situations that may prompt defensive medicine. Further research is needed to unveil the actual cost of defensive medicine in Jordan.

## Supporting information

**S1 File. DM survey.**
(PDF)

**S1 Table. Frequency of main causes of DM behaviors among physicians by sector.**
(DOCX)

**S2 Table. Average frequency score of DM behaviors during a typical working week by sector.**
(DOCX)

**S3 Table. Average frequency score of reasons why the practice of defensive medicine can be considered a favoring/limiting factor for the professional practice by sector.**
(DOCX)

**S4 Table. Average frequency score of reasons why the practice of defensive medicine can be considered a favoring/limiting factor for the patients by sector.**
(DOCX)

**S5 Table. Average effectiveness score of strategies in changing defensive medicine behaviors by sector.**
(DOCX)

## Author Contributions

**Conceptualization:** Qosay Al-Balas.

**Data curation:** Shoroq Altawalbeh, Ibtihal Ibrahim.

**Funding acquisition:** Qosay Al-Balas.

**Investigation:** Qosay Al-Balas, Carmela Rinaldi.

**Methodology:** Shoroq Altawalbeh, Carmela Rinaldi, Ibtihal Ibrahim.

**Project administration:** Qosay Al-Balas, Shoroq Altawalbeh, Carmela Rinaldi.

**Writing – original draft:** Ibtihal Ibrahim.

**Writing – review & editing:** Qosay Al-Balas, Shoroq Altawalbeh, Carmela Rinaldi.

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
