## [Decision Letter · Decision Letter 0]

11 Apr 2023

PONE-D-23-01241The practice of defensive medicine among Jordanian physicians: a cross sectional study.PLOS ONE

Dear Dr. Al-Balas,

Thank you for submitting your manuscript to PLOS ONE. After careful consideration, we feel that it has merit but does not fully meet PLOS ONE’s publication criteria as it currently stands. Therefore, we invite you to submit a revised version of the manuscript that addresses the points raised during the review process.

We look forward to receiving your revised manuscript.

Kind regards,

Alessandro Vittori, M.D.

Academic Editor

PLOS ONE

Journal Requirements:

https://journals.plos.org/plosone/s/file?id=ba62/PLOSOne_formatting_sample_title_authors_affiliations.pdf"

‘Al-Balas Q. Received the fund.

Grant No.: 20200641

Funder: Jordan University of Science & Technology, Deanship of Reserch.

https://www.just.edu.jo/Deanships/DeanshipofResearch/Pages/Default.aspx

No role played in the study“       

“No.”

Additional Editor Comments:

Dear Authors, The reviewers have filmed finished reviewing your manuscript. The comments are generally favourable, even if the manuscript is not publishable in this form. Please follow the directions of the reviewers.

Kind Regards

Reviewers' comments:

Reviewer's Responses to Questions

**Comments to the Author**

1. Is the manuscript technically sound, and do the data support the conclusions?

Reviewer #1: Partly

Reviewer #2: Yes

Reviewer #3: Yes

2. Has the statistical analysis been performed appropriately and rigorously? 

Reviewer #1: Yes

Reviewer #2: Yes

Reviewer #3: Yes

3. Have the authors made all data underlying the findings in their manuscript fully available?

Reviewer #1: No

Reviewer #2: Yes

Reviewer #3: Yes

4. Is the manuscript presented in an intelligible fashion and written in standard English?

Reviewer #1: Yes

Reviewer #2: Yes

Reviewer #3: Yes

5. Review Comments to the Author

Reviewer #1: Dear Respectable Authors

Thank you for considering a significant area of research related to defensive medicine among physicians. Your results are of interest but your manuscript needs revisions as follows;

- Please add the data collection period in the abstract and the main text.

- The sample size is small. What is your reason?

- The abstract section, please add a summary of all results in the results section (prevalence, reasons, and strategies).

- Add data regarding the validity and reliability of the questionnaire. How did you adopt the Panella questionnaire?

- How do you conclude about the cost? In my opinion, and based on your results, discussing cost is not a reasonable conclusion at all.

- The abstract's aim is not similar to the aim at the end of the introduction.

- Please report the results separately by the public and private sectors.

- Please remove: from headings and subheadings sections.

- What is your sampling method? Please add a section regarding sample size and sampling method.

- There is a conflict regarding working seating. Public and private? or private, government, and university-based?

- I can not access the figures because the format is Tiff and not presented to me.

- Your discussion is week. You must discuss all your results with the results of the other related research and state the similarity and differences with reasons. Please add more updated research in this field. For example, this is a good scoping review that you can benefit it in your discussion section:

The occurrence, types, reasons, and mitigation strategies of defensive medicine among physicians: a scoping review

- The number of references is low.

Cheers

Reviewer #2: In the current study, authors have examined the practice of defensive medicine among Jordanian physicians. The manuscript is well-written and worth publishing. I have some minor comments:

- In the Introduction, I suggest that authors add information about Jordan in terms of population and health sectors.

- In the Methods, authors should provide information on the sample size calculations, sampling strategy, and distribution of the sample according to targeted sites.

- The IRB approval number and the name of the IRB should be added to the Method section.

- Authors should identify which targeted places are considered governmental and which ones are private.

- In Table 1, subcategories should be arranged in an ascending manner. For example, age and experience. Same thing applies to some of the other tables.

- Figures were not available in the PDF file

Reviewer #3: This cross-sectional study was conducted to evaluate the prevalence of Defensive Medicine (DM) practice among physicians in Jordan. The study sheds light on the reasons behind DM practice and potential strategies to reduce it. It is interesting to note that unfavorable legislation was reported as the primary reason for DM practice, and laboratory exams for diagnostic purposes were found to be the most practiced behavior.

The study also highlights the concern of physicians about increasing patterns and costs associated with DM in the future. Overall, this study provides valuable information about the prevalence and predictors of DM practice in a developing country like Jordan.

The article is very well organized, concise enough and from where I am standing, clearly meets its goals. Congratulations on that. That's why I hope that in the future articles like this one can have some impact at higher levels.

I will only a few recommendations and all of them are minor English sentence making/vocabulary suggestions. For example, line 243 "Findings from this study shows".

Another minor suggestion is to think about the sections organization if there is a better way to organize the content of introduction and discussion. In this regard, I suggest to include and discuss relevant references such as doi: 10.1213/ANE.0000000000003395 (By examining a current group of closed malpractice claims in which obstetric anesthesiology was the primary defendant, the authors explored the correlation between contributing factors, patient injuries, and legal outcomes).; doi: 10.3390/healthcare9081012 (a more recent paper on litigations in Italy, see your reference number 11); doi: 10.1016/j.spinee.2022.08.012. Epub 2022 Aug 23.

(In a comprehensive review of legal claims reported in the United States over the past 70 years, researchers examined the risks of litigation associated with sports-related spinal injuries. They found that such injuries can be catastrophic, especially in collision sports like football, and can result in substantial legal claims. Negligent supervision and premises liability were reported as the most common legal claims, and the median payout for all cases was approximately $780,000.).

6. PLOS authors have the option to publish the peer review history of their article (what does this mean?). If published, this will include your full peer review and any attached files.

Reviewer #1: **Yes: **Morteza Arab-Zozani

Reviewer #2: **Yes: **Omar F. Khabour

Reviewer #3: No

---

## [Author Response · Author response to Decision Letter 0]

19 May 2023

Reviewers' comments:

Dear Reviewers, 

we really appreciate your time and efforts spent in reviewing this work and your efforts to optimize it to the level of the respected journal; below is the reply for each point you raised and we thank you again for your time.

All the best

5. Review Comments to the Author

Reviewer #1: Dear Respectable Authors

Thank you for considering a significant area of research related to defensive medicine among physicians. Your results are of interest but your manuscript needs revisions as follows;

- Please add the data collection period in the abstract and the main text.

Response: 

"The collection period was from Jan 2021 to June 2021."

The text is edited accordingly in the methods and abstract.

- The sample size is small. What is your reason?

Response: 

Indeed, we were limited in sample size, and recruiting physicians was challenging due to COVID-19 restrictions and the high workload of physicians in both public and private sectors. In addition, the nature of study idea made physicians more hesitant to participate even though physicians were informed that the study data will be kept anonymous and will never be used for judgmental purposes. 

This is already added in the last paragraph of the discussion section of the text (the limitations). 

- The abstract section, please add a summary of all results in the results section (prevalence, reasons, and strategies).

Response:

The results section in the abstract is edited accordingly.

- Add data regarding the validity and reliability of the questionnaire. How did you adopt the Panella questionnaire?

Response: 

As mentioned in the methods section, the survey was fully validated for the purpose of the study. The original survey (Panella survey) was developed by a multidisciplinary team with expertise in clinical and risk management in March 2014, and was pilot tested for validity on 25 physicians, with an inter-test interval of 15 days between the first and the second validation tests (Pearson’s r = 0.8). Permission was obtained from the corresponding author of the reference article.

In our study, practicality and clarity of the survey questions were further evaluated through a pilot study of about 15 physicians. Feedback from pilot study participants were utilized to improve the questionnaire as needed. 

The text is updated accordingly.

- How do you conclude about the cost? In my opinion, and based on your results, discussing cost is not a reasonable conclusion at all.

Response: 

In the current study, we evaluated participants believes about the impact of DM behaviors on the costs of healthcare interventions, and we found that the majority of participants (87.4%) believe that the adoption of DM behaviors increases the costs of healthcare interventions.

We agree with the reviewer that conclusion about costs is challenging in the current study as well as all studies investigating DM. The conclusion section is edited to focus on main results: 

 “Defensive medicine practice is common among Jordanian physicians with concerns about increasing pattern in the future.”

- The abstract's aim is not similar to the aim at the end of the introduction.

Response: 

The aims were modified accordingly, please see the main text. 

- Please report the results separately by the public and private sectors.

Response:

In responding to this comment, all main results in this study were reanalyzed separately by the public and private sectors and were added as supplementary tables in the supplementary file. All supplementary tables are cited in the text. 

- Please remove: from headings and subheadings sections.

Response: 

The text modified accordingly.

- What is your sampling method? Please add a section regarding sample size and sampling method.

Response:

 we used convenience sampling in the current study.

The Methods/ Study design and population section is edited accordingly: “A self-administered questionnaire was distributed in person for a convenience sample of physicians from both sectors.”

Indeed, this was a descriptive study in which we used convenience sampling to recruit physicians. We acknowledge that we were limited in sample size, and recruiting physicians was challenging due to COVID-19 restrictions and the high workload of physicians in both public and private sectors. In addition, the nature of study idea made physicians more hesitant to participate even though physicians were informed that the study data will be kept anonymous and will never be used for judgmental purposes. This issue is highlighted in the limitations:

- There is a conflict regarding working seating. Public and private? or private, government, and university-based?

Response:

 To clarify: Public sector hospitals included in the current study were:

- King Abdullah University Hospital (KAUH), a teaching hospital.

- Hospitals belong to Royal Medical Services (RMS) and Ministry of Health hospitals (MoH), government hospitals.

The Methods/ Study design and population section is edited accordingly: “Public sector hospitals included in the current study were King Abdullah University Hospital (KAUH) which is a teaching hospital, and government hospitals belong to Royal Medical Services (RMS) and Ministry of Health hospitals (MoH).”

- I can not access the figures because the format is Tiff and not presented to me.

Response: 

Figures will be inserted at the end of this reviewers' response.

- Your discussion is week. You must discuss all your results with the results of the other related research and state the similarity and differences with reasons. Please add more updated research in this field. For example, this is a good scoping review that you can benefit it in your discussion section:

The occurrence, types, reasons, and mitigation strategies of defensive medicine among physicians: a scoping review

Response: 

Thank you for your note, some of the text modified accordingly and the reference suggested was very informative and has been added

- The number of references is low.

Response: 

More references were added, please see the main text.

Reviewer #2: In the current study, authors have examined the practice of defensive medicine among Jordanian physicians. The manuscript is well-written and worth publishing. I have some minor comments:

- In the Introduction, I suggest that authors add information about Jordan in terms of population and health sectors.

Response: 

Thank you, the main text was modified accordingly. 

- In the Methods, authors should provide information on the sample size calculations, sampling strategy, and distribution of the sample according to targeted sites.

Response:

 Indeed, this was a descriptive study in which we used convenience sampling to recruit physicians. We acknowledge that we were limited in sample size, and recruiting physicians was challenging due to COVID-19 restrictions and the high workload of physicians in both public and private sectors. In addition, the nature of study idea made physicians more hesitant to participate even though physicians were informed that the study data will be kept anonymous and will never be used for judgmental purposes. This issue is highlighted in the limitations:

“Indeed, we were limited in sample size, and recruiting physicians was challenging due to COVID-19 restrictions and the high workload of physicians in both public and private sectors. In addition, the nature of study idea made physicians more hesitant to participate even though physicians were informed that the study data will be kept anonymous and will never be used for judgmental purposes. “

In addition, the Methods/ Study design and population section is edited accordingly: “A self-administered questionnaire was distributed in person for a convenience sample of physicians from both sectors.”

The distribution of the sample according to targeted sites is reported in table 1; 54.29 % of participants were from private sector and the rest were from the public sector.

- The IRB approval number and the name of the IRB should be added to the Method section.

Response: 

This was done as recommended, please see the main text (the Methods section). Thank you

- Authors should identify which targeted places are considered governmental and which ones are private.

Response: 

To clarify: Public sector hospitals included in the current study were:

- King Abdullah University Hospital (KAUH), a teaching hospital.

- Hospitals belong to Royal Medical Services (RMS) and Ministry of Health hospitals (MoH), government hospitals.

The Methods/ Study design and population section is edited accordingly: “Public sector hospitals included in the current study were King Abdullah University Hospital (KAUH) which is a teaching hospital, and government hospitals belong to Royal Medical Services (RMS) and Ministry of Health hospitals (MoH).”

- In Table 1, subcategories should be arranged in an ascending manner. For example, age and experience. Same thing applies to some of the other tables.

Response: 

Thank you, this was corrected accordingly.

- Figures were not available in the PDF file

Response: 

Figures will be inserted at the end of this reviewers' response.

Reviewer #3: This cross-sectional study was conducted to evaluate the prevalence of Defensive Medicine (DM) practice among physicians in Jordan. The study sheds light on the reasons behind DM practice and potential strategies to reduce it. It is interesting to note that unfavorable legislation was reported as the primary reason for DM practice, and laboratory exams for diagnostic purposes were found to be the most practiced behavior.

The study also highlights the concern of physicians about increasing patterns and costs associated with DM in the future. Overall, this study provides valuable information about the prevalence and predictors of DM practice in a developing country like Jordan.

The article is very well organized, concise enough and from where I am standing, clearly meets its goals. Congratulations on that. That's why I hope that in the future articles like this one can have some impact at higher levels.

I will only a few recommendations and all of them are minor English sentence making/vocabulary suggestions. For example, line 243 "Findings from this study shows".

Another minor suggestion is to think about the sections organization if there is a better way to organize the content of introduction and discussion. In this regard, I suggest to include and discuss relevant references such as doi: 10.1213/ANE.0000000000003395 (By examining a current group of closed malpractice claims in which obstetric anesthesiology was the primary defendant, the authors explored the correlation between contributing factors, patient injuries, and legal outcomes).; doi: 10.3390/healthcare9081012 (a more recent paper on litigations in Italy, see your reference number 11); doi: 10.1016/j.spinee.2022.08.012. Epub 2022 Aug 23.

(In a comprehensive review of legal claims reported in the United States over the past 70 years, researchers examined the risks of litigation associated with sports-related spinal injuries. They found that such injuries can be catastrophic, especially in collision sports like football, and can result in substantial legal claims. Negligent supervision and premises liability were reported as the most common legal claims, and the median payout for all cases was approximately $780,000.).

Response: 

All the mentioned references were introduced and discussed in the main text. Please see the main text.

Thank you

Figures:

Figure 1

Figure 2

Figure 3

---

## [Decision Letter · Decision Letter 1]

18 Jul 2023

The practice of defensive medicine among Jordanian physicians: a cross sectional study.

PONE-D-23-01241R1

Dear Dr. Qosay Al-Balas,

We’re pleased to inform you that your manuscript has been judged scientifically suitable for publication and will be formally accepted for publication once it meets all outstanding technical requirements.

Kind regards,

Alessandro Vittori, M.D.

Academic Editor

PLOS ONE

Additional Editor Comments (optional):

I am pleased to inform you that your manuscript may be accepted.

Kind Regards

Reviewers' comments:

Reviewer's Responses to Questions

**Comments to the Author**

1. If the authors have adequately addressed your comments raised in a previous round of review and you feel that this manuscript is now acceptable for publication, you may indicate that here to bypass the “Comments to the Author” section, enter your conflict of interest statement in the “Confidential to Editor” section, and submit your "Accept" recommendation.

Reviewer #1: All comments have been addressed

Reviewer #2: All comments have been addressed

2. Is the manuscript technically sound, and do the data support the conclusions?

Reviewer #1: Yes

Reviewer #2: Yes

3. Has the statistical analysis been performed appropriately and rigorously? 

Reviewer #1: Yes

Reviewer #2: Yes

4. Have the authors made all data underlying the findings in their manuscript fully available?

Reviewer #1: Yes

Reviewer #2: Yes

5. Is the manuscript presented in an intelligible fashion and written in standard English?

Reviewer #1: Yes

Reviewer #2: Yes

6. Review Comments to the Author

Reviewer #1: Dear Respectable Authors

Thank you for your clarification. Your manuscript is acceptable in this fashion.

Cheers

Reviewer #2: Authors have answered my comments. I am satisfied with the revision. I recommend acceptance of the manuscript.

7. PLOS authors have the option to publish the peer review history of their article (what does this mean?). If published, this will include your full peer review and any attached files.

Reviewer #1: **Yes: **Morteza Arab-Zozani

Reviewer #2: No

---

## [Editor Report · Acceptance letter]

27 Oct 2023

PONE-D-23-01241R1 

The Practice of Defensive Medicine Among Jordanian Physicians: A Cross Sectional Study. 

Dear Dr. Al-Balas:

I'm pleased to inform you that your manuscript has been deemed suitable for publication in PLOS ONE. Congratulations! Your manuscript is now with our production department. 

Kind regards, 

on behalf of

Dr. Alessandro Vittori 

Academic Editor

PLOS ONE